

# Storm surge forecasting: quantifying errors arising from the double-counting of radiational tides.

Joanne Williams[1], Maialen Irazoqui Apecechea[2], Andrew Saulter[3], and Kevin J. Horsburgh[1]

[1]National Oceanography Centre, Joseph Proudman Building, 6 Brownlow St, Liverpool, UK
[2]Deltares, Delft, Netherlands
[3]Met Office, Fitzroy Road, Exeter, UK

*Correspondence to:* JOANNE WILLIAMS (JOLL@NOC.AC.UK)

**Abstract.** Tide predictions based on tide-gauge observations are not just the astronomical tides, they also contain radiational tides - periodic sea level changes due to atmospheric conditions and solar forcing. This poses a problem of double-counting for operational forecasts of total water level during storm surges. In some surge forecasting, a regional model is run as tide-only, with astronomic forcing alone; and tide-and-surge, forced additionally by surface winds and pressure. The surge residual is

defined to be the difference between these configurations and is added to the local harmonic predictions from gauges. Here we use the Global Tide and Surge Model based on Delft-FM to investigate this in the UK and elsewhere, quantifying the weather-related tides that may be double-counted in operational forecasts. We show that the global $S_2$ atmospheric tide is captured by the tide-surge model, and observe changes in other key constituents, including $M_2$. We also quantify the extent to which the "Highest Astronomical Tide", which is derived from tide predictions based on observations, may contain weather-related

components.

## 1   Introduction

The operational forecast in several countries of storm surge still-water levels is based on a combination of a harmonic tidal prediction and a model derived forecast of the meteorologically induced storm surge component. In the UK, the forecast is based on the "non-tidal residual", the difference of two model runs with and without weather effects. This is linearly added

to the "astronomical prediction" derived from local tide gauge harmonics (Flowerdew et al., 2010). This approach is taken because the complexity and large range of the UK tides is such that it has historically been difficult to model them to sufficient accuracy. The same method was applied in the Netherlands until 2015 when improvements to the local surge model DCSM-v6 made it unnecessary. It is still in use operationally in the extra-tropical US, where results of the SLOSH surge model are added to local tidal predictions (National Weather Service, 2018); similarly in Germany, using the BSHsmod model (BSH, 2018);

and is also used in the new Aggregate Sea-level Forecasting under evaluation in Australia, which also incorporates sea-level anomalies from a global baroclinic model (Taylor and Brassington, 2017).

There are several possible sources of error in this procedure. The purpose of the combined tide-and-surge model is to capture the well-documented non-linear interactions of the tide and surge. (e.g. Proudman, 1955). Yet the forecasting procedure assumes that the non-tidal residual may be added linearly to a gauge-based tide prediction. There is also an assumption that the





tide-only model and the harmonic prediction from the tide gauge are equivalent. In fact, the harmonics at the gauge will also be affected by the weather. There is therefore the potential for double-counting of radiational (weather-related) tidal constituents.

Specific radiational tides have been studied using response analysis, for example the solar-diurnal $\mathbf{S}_1$ by Ray and Egbert (2004), and semi-diurnal $\mathbf{S}_2$ by Dobslaw and Thomas (2005). Here we use the Global Tide and Surge Model (GTSM) based on Delft-FM, to compare many constituents and their combined effect globally. We find that the double counting of radiational tides has a potential contribution not just on long time scales (through $\mathbf{S}_a$, $\mathbf{S}_{sa}$) but also on a fortnightly cycle due to variations in $\mathbf{S}_2$ and in the phase of $\mathbf{M}_2$. We demonstrate the atmospheric tide at $\mathbf{S}_2$ may be observed in the GTSM model. We also show that the assumption of non-linearity may introduce errors if phase predictions disagree between model and observations.

We also quantify the extent to which the Highest and Lowest Astronomical Tide (HAT and LAT) are influenced by weather-related tides, and show that in many places several cm of what is reported as HAT is attributable to periodic weather patterns.

## 2 Surge forecasting

The harmonic analysis function (Pugh and Woodworth, 2014, Chapter 4) gives the tide prediction $H$ as:

$$H(t) = Z_0 + \sum_N A_n f_n \cos\left[\sigma_n t - g_n + (V_n + u_n)\right], \tag{1}$$

where $Z_0$ is the mean of the gauge data and the amplitudes $A_n$ and phases $g_n$ are associated with the tidal constituents with astronomically-determined frequencies $\sigma_n$. $f_n(t)$ and $u_n(t)$ are nodal adjustments to amplitude and phase, applied in order to allow for the 18.61 year nodal cycle and 8.85 year longitude of lunar perigee cycle. $V_n$ are the phases of the equilibrium tide, which we take as for Greenwich. We use UTC for all times to enable consistency between local gauges and global maps.

The current procedure for forecasting total water level in the UK is as follows:

1. Run a barotropic shelf model (CS3X, currently transitioning to NEMO Surge (O'Neill and Saulter, 2017)) in surge-and-tide mode, forced by an ensemble of wind and pressure from the current weather forecast to give timeseries $M_s(\boldsymbol{x},t)$ at each location $\boldsymbol{x}$. Also run the shelf model in tide-only mode, to get $M_t(\boldsymbol{x},t)$. Get the residual from these models, $M_r = M_s - M_t$.

2. Find the amplitude $A_n$ and phases $g_n$ of harmonics at each individual gauge location from past tide records. Derive a tide prediction $H_g(\boldsymbol{x}_g,t)$ from the gauge harmonics.

3. Forecast the total water level $W_f$ as model residual plus gauge harmonic prediction, $W_f(\boldsymbol{x}_g,t) = M_r(\boldsymbol{x}_g,t) + H_g(\boldsymbol{x}_g,t)$.

4. Finally, it has been proposed (Hibbert et al., 2015) that the forecast could apply various "empirical corrections" to nudge the forecast towards the observed level $W_g$ based on the mismatch of the peak tide over the last few days.

### 2.1 Selection of tidal constituents

The choice of tidal constituents used for the harmonic analysis varies according to the length of data available. We use 62 harmonics where there is one year's data, 115 for more than one year, as listed in table C1. To derive harmonics from the





global model (discussed below), from only 1 month's data, we use 26 independent primary constituents, and a further 8 related constituents.

## 2.2 Quantifying the effect on forecast of double-counting radiational tides

A significant source of error for this method is that a gauge is measuring the total water level, and hence the harmonic prediction

$H_g$ includes all wave, steric and surge effects. This is not therefore a prediction of the astronomical tide alone. Steric and wave effects are omitted by the model, but $M_s$ does include periodic radiational effects, which may be double-counted. We can test a minimum effect of this double-counting purely within the model by using $H_s$, the harmonic prediction of the model *including surge*, as a proxy for the harmonics of the observations at gauges. Then the forecast procedure can be estimated as $M_r + H_s$.

To estimate the error in this model forecast we can once again use the model, assuming $M_s \approx W_g$. Hence error $= M_s -$

$(M_r + H_s) = M_t - H_s$. That is, the minimum error from the current forecast procedure is equal to the error in the harmonic prediction from the model including surge at estimating the *tide-only* model, figure 1(top). There are several striking features here, annual cycles peaking around March in the Arctic, January in South-East Asia, and June in Europe. Fortnightly cycles occur almost everywhere, with amplitudes of several cm. We will examine the causes of these below.

If it were possible to avoid the double-counting, and provide astronomical tidal harmonics $H_t$ for the observations, this would

instead be equivalent to $M_r + H_t$, and the error would become $M_s - (M_r + H_t) = M_t - H_t$, as shown in figure 1(bottom). Since we're using the model as proxy for observations, if the harmonic prediction is an exact reproduction of the tide-only model then this would be exact. It is less than 5 cm at most UK sites and the monthly cycle has gone, but in the Bristol Channel there is still an error of up to around 0.5 m, indicating that the selection of 62 harmonic constituents are not capturing all of the model tide. This is consistent with Flowerdew et al. (2010) who found an "average [across UK ports] rms error of 7 cm with a

maximum value of 29 cm at Newport, in the Bristol Channel", using 50 constituents on the C3SX model.



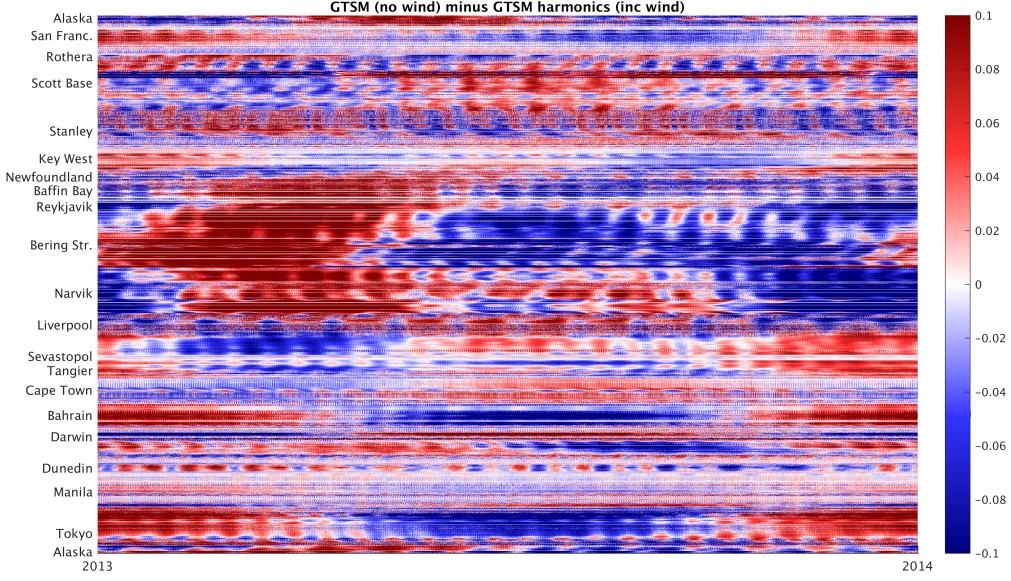

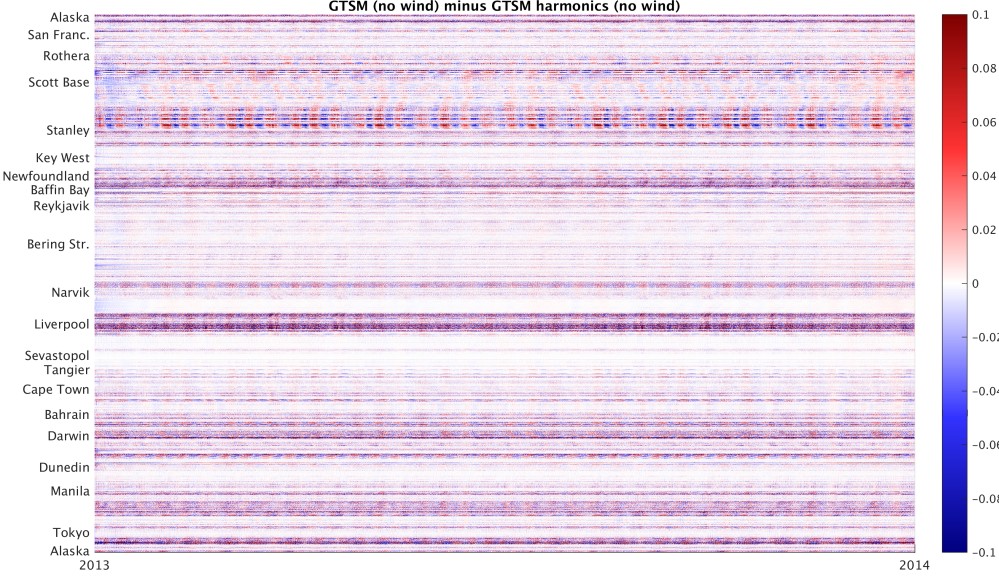

**Figure 1.** Time series of error (m) in harmonic prediction with 62 constituents (top panel) including surge and (bottom panel) tide-only at estimating the tide-only model, GTSM 2013 only. See appendix B for explanation of vertical coastal axis.





## 2.3 Fortnightly cycle arising from small changes to $S_2$ phase

$M_2$ has a period of 12.42 hours and $S_2$ exactly 12 hours. Through a lunar month they gradually move in and out of phase with each other, resulting in a spring-neap cycle. A small change in phase to $S_2$ harmonic would result in a change of which days it is in phase with $M_2$, and hence a substantial change in total tidal amplitude at a given date. For example, $S_2$ in the GSTM
model with/without surge has an amplitude change in the Bristol Channel of about 0.04 m, however there is a phase change of around $3°$ (7 min). Figure 2 shows the total change in amplitude at Avonmouth varies between the tide-only and surge models by up to 5–8 cm (on a fortnightly cycle between these limits) from this effect alone. This cycle could account for that seen by Byrne et al. (2017), and that in figure 2 of Flowerdew et al. (2010).

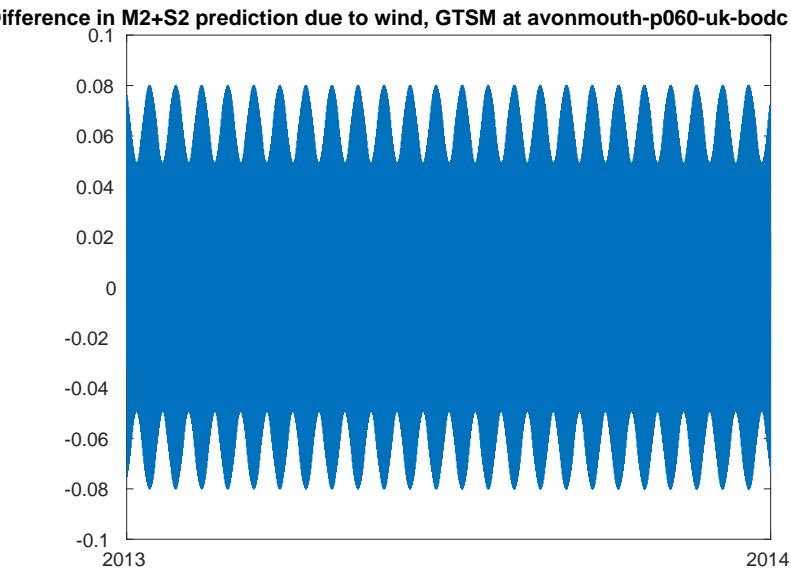

**Figure 2.** Fortnightly cycle of prediction change (metres) due to small changes in constituents $M_2$+$S_2$ alone, between model with/without surge. Avonmouth $S_2$ amplitude change = 3.5 cm, phase change $3.5°$, $M_2$ amplitude change 1 cm, phase change $0.2°$.

## 2.4 Quantifying surge-forecasting error due to disregarding non-linearity

The approach of linear addition of the harmonic prediction to the non-linear residual, $W_f = M_r + H_g$, itself carries a risk of error, even if the harmonic prediction did not include any radiational forcing.

Consider the following simplified example. Suppose the model tide has an $M_2$ amplitude of 3 m (ignore all other constituents), and there is a surge which has the effect of adding a constant amplitude 0.2 m and advancing the tide by a constant 30 min. Suppose also that the harmonics of the observed tide have the same amplitude as the tide-only model, but are out of
phase by 5 min (equivalent to $2.4°$ phase change for $M_2$). As illustrated in figure 3, the residual $M_r$ is decreasing during High Water due to the advanced tide. So if the observed harmonics have High Water later than the model, the forecast skew surge



is underestimated by 3 cm. If the observed harmonics predict High Water earlier than the model, the forecast skew surge is overestimated by 3 cm.

This example illustrates the importance of accurate phase agreement between model and observations, as well as amplitude.

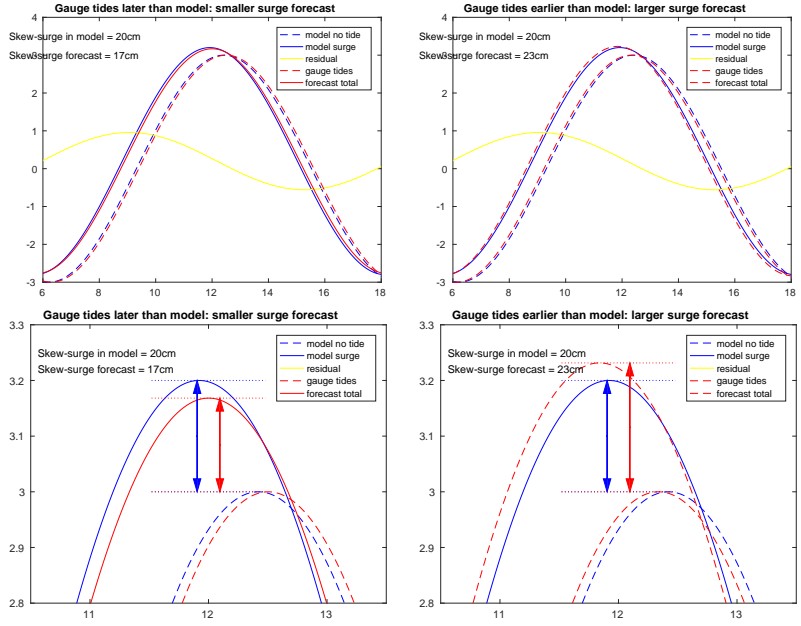

**Figure 3.** A surge is imposed which adds a constant amplitude of 20 cm and advances the tide by a constant 30 min. If the harmonics of the observations differ in phase by 5 min from the model a forecast error of $\pm 3$ cm will result as shown. Lower panels are magnified to show the high water.

# 3 The difference of specific harmonics

Figure 4 shows the vector difference in individual constituents between surge and tide-only models run for 2013, along the coast globally. With some exceptions in the Arctic and Antarctic, the effect on $S_a$ is around 5–20 cm, with around half that effect on $S_{sa}$, although there is an annual effect only in South-East Asia. $MS_f$ is affected by the surge component, as a side effect of the interaction between $M_2$ and $S_2$. This is because $MS_f$ is the fortnightly constituent which arises from the combination of $M_2$ and $S_2$, with a speed equal to the difference of their speeds. $MS_4$ is the counterpart to this, with a speed equal to the sum of the speeds of $M_2$ and $S_2$ (Pugh and Woodworth, 2014). Less explicable is effect on $M_m$ and $M_f$, but it may be due to insufficient separation with $MS_f$ over a relatively short record. The diurnal constituents $K_1$ and $O_1$ are affected by less than 5 cm, and are only changed regionally in the Antarctic. $S_1$ however is everywhere less than 1 mm in the tide-only model, but with the surge model peaks at 5 mm in northern Australis, the broadest regional effect being has 2–3 mm, in South-East Asia, consistent with the findings of (Ray and Egbert, 2004).





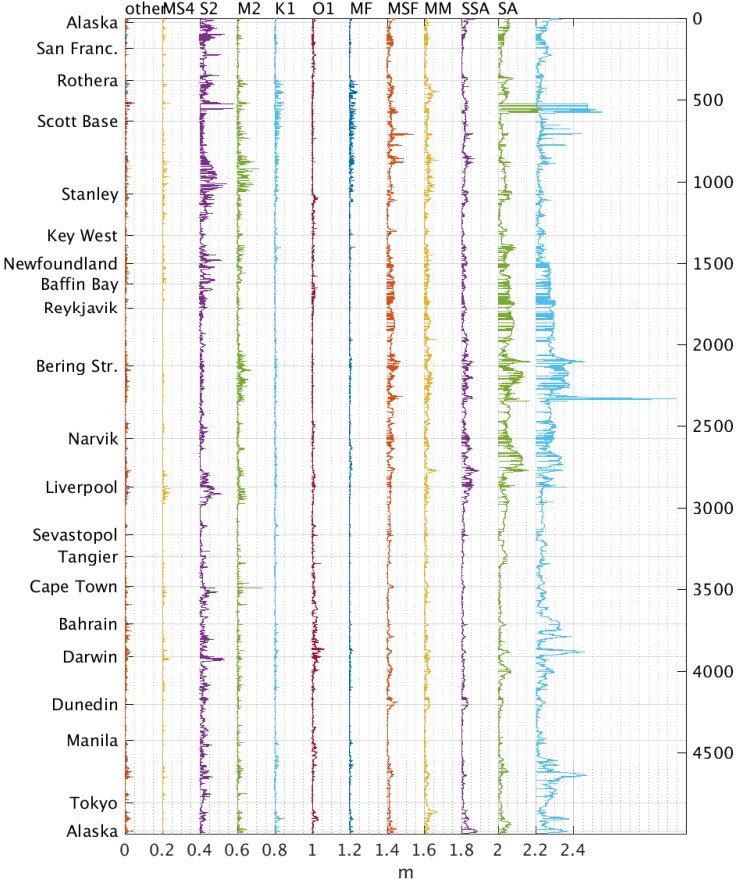

**Figure 4.** Vector difference (m, offset) between coefficients fitted to model including surge or tide only, GTSM 2013 only, 62 constituents fitted. This is the breakdown into constituents of the the difference between panels of figure 1. The maximum effect for these harmonics and others are given in table C1. See appendix B for explanation of coastal axis.

It may come as a surprise that constituents such as $M_2$, which has a purely lunar frequency, could possibly be effected by the weather. But this is where the non-linear interaction of surge and tide comes into play. The surge may consistently advance the phase of the tide during low pressure events and certain wind configurations. A high pressure system could delay the phase of the tide, but there is asymmetry between these events, so there is a net bias on the phase when the weather is included.

5    The effect on higher order constituents is everywhere less than 5 cm. The maximum difference in the UK and globally for each constituent is given in Appendix C. In the UK, the constituents affected the most by including the surge are $S_2$, $S_{sa}$, $M_2$, $S_a$, $M_m$, $MS_4$, $MS_f$ and $M_f$, with a maximum change of $> 2$cm, and a further 19 constituents change 1–2 cm somewhere on

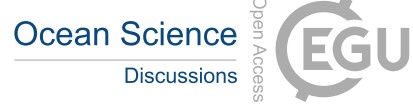

the UK coast. Globally, $S_a$ and $S_{sa}$ are far more significant, but $S_2$, $M_f$, $M_2$, $M_m$, $MS_f$, $S_1$, $K_1$, $K_2$, $O_1$, $M_{A2}$, and $MS_4$ all change more than 4 cm (somewhere on the global coast). Vector differences of 13 cm in $S_2$ is seen in north-west Australia.

These results are robust to the number of constituents fitted (115, 62 or 34) to within 2 mm.



## 3.1   S$_2$ atmospheric tide

Some of the difference between harmonics of surge and tide-only models is directly attributable to the atmospheric tides. The atmospheric pressure has S$_2$ variations with amplitude of about $1.25\cos^3\phi$ millibars, for latitude $\phi$ (Pugh and Woodworth, 2014). The ocean response to the ERA Interim forcing at S$_2$ is contained in the difference between harmonic predictions of the model runs with and without surge (figure 5). It is consistent with response analysis based on the S$_2$ tides seen in ECMWF reanalysis data (figure 2, Dobslaw and Thomas, 2005), and in a 2-layer model forced by 8 constituents (figure 1b, Arbic, 2005).

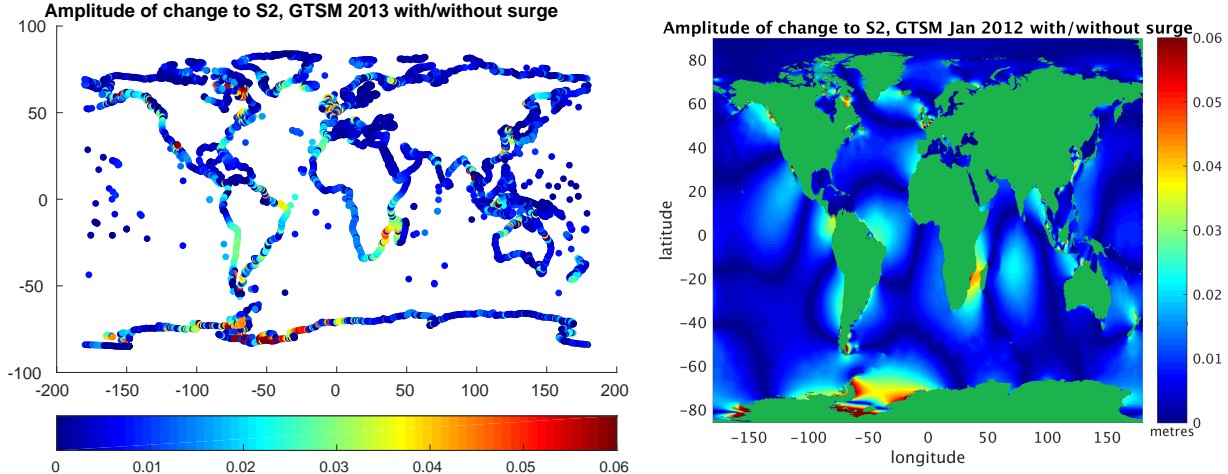

**Figure 5.** Amplitude (m) of S$_2$ difference between between coefficients fitted to GTSM model including surge or tide only. First panel: coastal data only, whole of 2013; second panel: 26 primary coefficients fitted to January 2012 only.





## 4  Highest Astronomical Tide

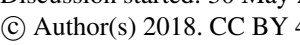

**Figure 6.** (a) Range calculated from max & min of 18.6yr reconstruction, 15 min interval, from 26 primary and 8 related constituents derived from 1 month tide-only GTSM, and the nodal tide. (b) Difference between (a) and $2(M_2 + S_2 + O_1 + K_1)$, from the same run. (c) Extreme tides LAT (blue) and HAT(red, offset 1 m), difference between model including surge or tide only, derived from 62 constituents from GTSM 2013. See appendix B for explanation of coastal axis.





The Highest Astronomical Tide (HAT) is used internationally for flood-forecasting references levels and in navigation for clearance under bridges. HAT can be used in structural design with skew surge as an independent variable for determining return period water levels. Lowest Astronomical Tide (LAT) is also an important parameter, recommended for use as the datum on navigation charts (IHO, 2017). Once the phases and amplitudes $A_n$ and $g_n$ are known, $H(t)$ is fully determined for all time

by equation (1), and the future HAT and LAT are given by $\max(H(t))$ and $\min(H(t))$. But because of the overlap in phase of the forcing between the constituents, and the $f_n$ and $u_n$ nodal adjustments, it is not trivial to write HAT or LAT algebraically. They are therefore determined by inspection of the predicted tides, preferably over a 18.6 year nodal cycle. Figure 6a shows the range, HAT minus LAT, when we do this by synthesising a predicted tide at 15 minute intervals over 18.6 years, globally.

The quick calculation of Range $= 2(\mathbf{M}_2 + \mathbf{S}_2 + \mathbf{O}_1 + \mathbf{K}_1)$ is occasionally used (e.g. Yotsukuri et al., 2017), but the error due

to this can be over 1 metre (figure 6b), and although in the open ocean most of this is due to omission of the nodal tide, near the coast up to 0.5 m of the maximum range can be due to the shallow water constituents. Radiational effects are omitted from figure 6a, which is based on a tide-only run. Also, this analysis was limited to 1 month, so uses only 34 constituents, therefore omitting $\mathbf{S}_1$ and the long-period contributions to HAT and LAT.

Figure 6c shows the effect on HAT and LAT using the constituents derived with surge in the GTSM. In most of the UK,

the HAT goes down when the surge model is used to generate the tidal predictions, rather than going up. This is because the peak of the weather-related components does not coincide with the maximum astronomical effects alone. This implies that since the tide-gauge predictions include surge, the observation-based HAT in the UK is actually about 10 cm lower than true astronomical-only tidal height. But in many places round the world the HAT is higher when the surge model is used. So the observation-based HAT actually has been raised by some radiational component.

LAT tends to move the opposite way, so in most places the maximum tidal range is increased by using the tide+surge model. That is, the true astronomical-only tidal range is slightly less than that quoted from harmonics based on predictions. In Scotland, (just above Liverpool in figure 6c) both LAT and HAT go down when the surge model is used to generate the tidal predictions, so the quoted LAT and HAT is actually about 10cm lower than astronomical only.

The most extreme changes shown in Figure 6c are in the Arctic and Antarctic, and should be interpreted with some caution

as these areas are the least well understood in the model.

In places with small tide, seasonal signals may be dominant and they may be important to include for practical purposes. For example along the French/Italian coast from Mallorca to Sicily there's about a 7cm increase in HAT and 3cm decrease in LAT using the surge rather than tide-only model, so a Highest "Astronomical" Tide based on predicted tide from observations actually contains about 7 cm due to seasonal winds.



## 5 Conclusions

There are substantial changes in tidal constituents fitted to tide-only and tide-and-surge model results. Even constituents with purely lunar frequencies, including $M_2$, may be affected by the surge, perhaps owing to asymmetry in phase changes of the tide under high and low pressure weather systems.

Some effects of the weather on tides are double-counted in the forecast procedure used in the UK, where model residuals are added to gauge-based tide predictions. Even if the model were perfect, the minimum error from the current forecast procedure would be at least the error in the harmonic prediction including surge at estimating the tide-only model. If 62 constituents are fitted, this has a standard deviation of 20 cm at Avonmouth and 4–10 cm at most other UK gauges. 5–8 cm of the error at Avonmouth is due simply to a small change in phase of the $S_2$ harmonic. Further errors in total water level and skew surge

arise directly from the linear addition of the harmonic prediction to the non-linear residual, particularly where there is a phase difference between model and gauge tidal harmonics.

    Understanding and quantifying these errors is extremely important for forecasters, who will often need to advise or intervene on the expected surge risk, often based on a direct comparison between observed residuals and the forecast non-tidal residual. Where, for example, such a comparison may lead to the observed residual falling outside the bounds of an ensemble of forecast

non-tidal residuals, the forecaster may significantly (and potentially incorrectly) reduce their confidence in the model's estimate of surge if they are unaware of the additional errors associated with the harmonic tide and whether or not they have been addressed within the ensemble forecast's post processing system. For comparison, across the UK-wide set of Class A ports short range ensemble forecast RMS spread is of order 5–10 cm (Flowerdew et al., 2013). It is noted that, in the UK, the majority of coastal flood events occur around peak spring tides (Haigh et al., 2015), where the sensitivity to any errors in the $M_2$–$S_2$

phase relationship is arguably at its highest.

    The atmospheric tide, at $S_2$ is present in the ERA Interim forcing, and the ocean response to it, with amplitude about 1–5 cm, can be seen in the difference between the model results with and without surge. There is therefore an argument for including an atmospheric tide forcing in a "tide-only" model, and this is being explored by (Irazoqui Apecechea et al., 2018 *(In review)*). In this case, care would need to be taken to omit the direct atmospheric tide forcing in the tide+surge version, to avoid a different

form of double-counting.

    The estimates of Highest and Lowest Astronomical Tide are influenced by radiational tides. HAT and LAT are most readily calculated by inspecting long time series of predicted tides, and if observation-based, these predictions will include weather-related components. In most places globally this results in HAT being calculated as higher than the strictly astronomical component, and LAT being lower, however the opposite is true in the UK. The effects are of the order of ~10 cm.

For many practical purposes it is correct to include predictable seasonal and daily weather-related cycles in the HAT and LAT. However the separate effects should be understood, as the radiational constituents may be subject to changing weather patterns due to climate change. It is also important not to double-count weather effects, if HAT or LAT are used in combinations with surge for estimating return-period water levels.

    These considerations about HAT would also apply (proportionally less) to other key metrics such as mean high water.



## Appendix A:  Global Tide and Surge Model

GTSM is the forward Global Tide and Surge Model developed at Deltares on a base of Delft-FM (Flexible Mesh) (Verlaan et al., 2015.; Irazoqui Apecechea et al., 2018 *(In review)*). The model is under active development, and the version used in this paper is has resolution from around 50 km in the open ocean to around 5 km at the coast. The atmospheric forcing is the ECMWF

ERA-Interim 6-hourly reanalysis (Dee et al., 2011), downloaded at $0.25°$ resolution but from a spherical harmonic equivalent to $\tilde{0}.75°$. Validation of the major tidal coefficients has been favourable, and although the surge forecast under-predicts the effect of major hurricanes, due to lack of resolution in the weather forecast, most surge events are captured. For this report a 2013 run is used, with an 11 day spin-up period in December 2012. 62 tidal constituents are found by harmonic analysis of the 2013 results.

## Appendix B:  Ordering of model sites around the coast

Due to limitations of data storage the model is only output at high frequency at all grid points for one month (Jan 2012) and a selection of coastal points for the year 2013. These coastal points are spaced roughly every 80 km, and also wherever a tide gauge is situated. Due to automatic procedures to select output sites, a few may be incorrectly sited at model dry sites - these are clearly seen in plots as lacking sufficient high-frequency variability. The along-coast plots are ordered approximately

anti-clockwise around the UK including neighbouring coasts in Europe and Ireland. The order is indicated in figure B1.

The algorithm for coastal order is as follows:

1) Define a single global coastline polygon.

This is done using the GSHHG (Global Self-consistent, Hierarchical, High-resolution Geography) data set (Wessel and Smith, 1996), version gshhg2.3.6 (August 19, 2016, downloaded from www.ngdc.noaa.gov) . We use the coarse resolution,

with only Level 1 (coastline) and Level 6 (Antarctic Ice Shelf), although consistent results for this technique can be obtained including enclosed lakes. To merge the separate landmasses and islands into a weakly simple polygon, topologically equivalent to a disc, we start with a single landmass and add others in turn using pairs of identical edges as "bridges". We start with the main landmass of Eurasia $L_1$, and find the closest vertex $l$ to a vertex $p$ from any of the remaining polygons $[P_2,...P_N]$. Suppose $p$ belongs to polygon $P_j$. Then we add $P_j$ to $L_1$ using two new edges $\overrightarrow{lp}$ and $\overrightarrow{pl}$, to give a new merged polygon $L_2$.

The vertices of $L_2$ are then $[L_1(1:l), P_j(p:end, 1:p-1), L_1(l:end)]$. Now repeat, searching for the nearest point in $L_2$ to any vertex in the remaining polygons $[P_2,...,P_{j-1}, P_{j+1},...P_N]$. It is necessary for all initial polygons to be defined in the same sense (anticlockwise). If inland seas (Level 2) are included, they should be defined clockwise. The GSHHG data is consistent with this definition. Distance for nearest points are defined as arc-length on a sphere.

This technique has the benefit of tending to group island chains together in a consistent order. It cannot produce crossing

edges. Because polygons are added in distance order, islands near continents are added to their neighbouring coast, and remote mid-ocean islands tend to be clustered, and attached to the nearest continent. The coasts of the Pacific, Atlantic and Indian and Arctic Oceans are all treated clockwise. Antarctica is attached across Drake Passage and ordered Westward. Nearby locations across narrow islands (particularly Sumatra), isthmuses (Panama), and straits (Gibraltar) may be widely separated in the order.



But neighbouring points in the order can be expected to have fairly smoothly varying oceanography, with the "bridges" often, although not necessarily, approximating shoals.

As a final step we adjust the starting point of $L_2$ to be in Alaska, for convenience of mapping.

2) Rank the coastal points according to the nearest point on the global polygon.

5    Having defined this coastal order, we can apply it to any coastal data set, for example tide gauges. We number the vertices $[1, ..., K]$. For each of the tide gauge locations $T$ we find the nearest vertex $k$, and then rank the gauges according to $T_k$. In the event of gauges being much closer than the resolution of the vertices, a quick method for refinement is to linearly interpolate with extra vertices along polygon edges. Some problems may also occur with islands not in the coarse resolution data, which will tend to jump to the nearest coast.

10    A further advantage here is that having defined the coastal polygon, the same order can be applied to different data sets and models, leading to closely comparable along-coast plots.

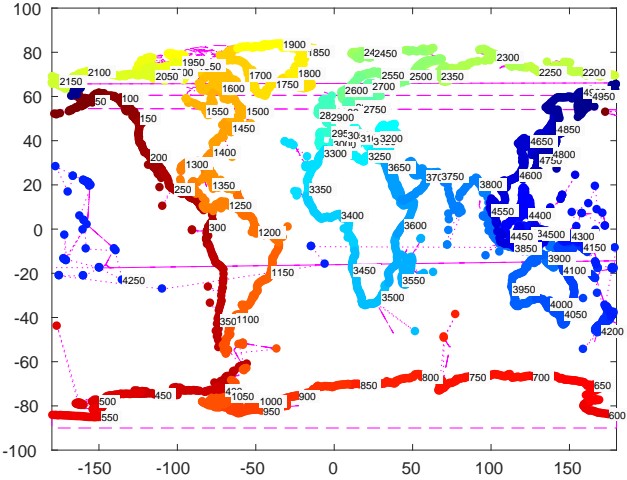

**Figure B1.** Sites used for analysis and coastal ordering (Red to Blue is top-to-bottom of other plots)





## Appendix C: Tidal constituents

Table C1 lists the constituents used in this paper. For short records, related constituents are used, and we fit 34 constituents with only 26 independent terms. We follow the usual convention of $a$ for annual, $f$ fortnightly, 2 for approximately semi-diurnal etc., which are given as subscripts in the main text.

**Table C1.** Tidal Harmonic Constituents referred to in this paper, and the maximum change between models run tide-only or with surge forcing, at coastal locations, as from figure 4.

| Name | Speed (deg/hour) | 18yr list | 1 year list | Max effect of surge in UK(cm) | Max effect of surge globally (cm) | 1 month list Related | 1 month list Primary |
|---|---|---|---|---|---|---|---|
| Total: | | 115 | 62 | | | 8 | 26 |
| Sa | 0.041 069 | yes | yes | 4.8 | 74.8 | - | - |
| Ssa | 0.082 137 | yes | yes | 5.6 | 23.4 | - | - |
| Mm | 0.544 375 | yes | yes | 4.2 | 9.3 | - | yes |
| MSf | 1.015 896 | yes | yes | 3.2 | 7.9 | - | yes |
| Mf | 1.098 033 | yes | yes | 2.1 | 14.0 | - | - |
| 2Q1 | 12.854 286 | yes | yes | 1.1 | 2.2 | - | - |
| sigma1 | 12.927 140 | yes | yes | 1.1 | 1.3 | - | - |
| Q1 | 13.398 661 | yes | yes | 0.7 | 1.6 | - | yes |
| rho1 | 13.471 515 | yes | yes | 0.7 | 1.2 | - | - |
| O1 | 13.943 036 | yes | yes | 0.7 | 4.3 | - | yes |
| MP1 | 14.025 173 | yes | yes | 0.6 | 1.5 | - | - |
| M1 | 14.496 694 | yes | yes | 0.5 | 1.4 | - | yes |
| chi1 | 14.569 548 | yes | yes | 0.3 | 1.0 | - | - |
| pi1 | 14.917 865 | yes | yes | 0.5 | 1.9 | yes | - |
| P1 | 14.958 931 | yes | yes | 0.9 | 3.1 | yes | - |
| S1 | 15.000 000 | yes | yes | 1.4 | 6.5 | - | - |
| K1 | 15.041 069 | yes | yes | 1.0 | 5.1 | - | yes |
| psi1 | 15.082 135 | yes | yes | 0.3 | 3.2 | yes | - |
| phi1 | 15.123 206 | yes | yes | 0.6 | 1.3 | yes | - |
| theta1 | 15.512 590 | yes | yes | 0.5 | 1.1 | - | - |
| J1 | 15.585 443 | yes | yes | 1.0 | 1.2 | - | yes |
| SO1 | 16.056 964 | yes | yes | 0.5 | 2.3 | - | - |
| OO1 | 16.139 102 | yes | yes | 0.5 | 1.4 | - | yes |

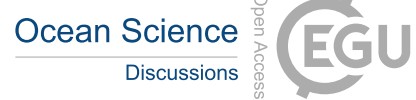

| Name | Speed (deg/hour) | 18yr list | 1 year list | Max effect of surge in UK(cm) | Max effect of surge globally (cm) | 1 month list Related | 1 month list Primary |
|---|---|---|---|---|---|---|---|
| Total: | | 115 | 62 | | | 8 | 26 |
| 2MN2S2 | 26.407 938 | yes | - | - | | - | - |
| 3M(SK)2 | 26.870 175 | yes | - | - | | - | - |
| 3M2S2 | 26.952 313 | yes | - | - | | - | - |
| OQ2 | 27.341 696 | yes | yes | 0.5 | 1.1 | - | - |
| MNS2 | 27.423 834 | yes | yes | 0.8 | 1.1 | - | - |
| MnuS2 | 27.496 687 | yes | - | - | | - | - |
| MNK2S2 | 27.505 971 | yes | - | - | | - | - |
| 2MK2 | 27.886 071 | yes | - | - | | - | - |
| 2N2 | 27.895 355 | yes | yes | 0.8 | 1.3 | yes | - |
| mu2 | 27.968 208 | yes | yes | 1.7 | 2.7 | - | yes |
| SNK2 | 28.357 592 | yes | - | - | | - | - |
| NA2 | 28.398 661 | yes | - | - | | - | - |
| N2 | 28.439 730 | yes | yes | 1.3 | 3.2 | - | yes |
| NB2 | 28.480 796 | yes | - | - | | - | - |
| nu2 | 28.512 583 | yes | yes | 0.8 | 1.5 | yes | - |
| OP2 | 28.901 967 | yes | yes | 1.2 | 2.8 | - | - |
| MA2 | 28.943 036 | yes | yes | 0.8 | 4.3 | - | - |
| M2 | 28.984 104 | yes | yes | 5.1 | 13.0 | - | yes |
| MB2 | 29.025 173 | yes | yes | 1.3 | 3.9 | - | - |
| MKS2 | 29.066 242 | yes | yes | 1.6 | 3.1 | - | - |
| lambda2 | 29.455 625 | yes | yes | 1.3 | 1.6 | - | - |
| L2 | 29.528 479 | yes | yes | 1.1 | 1.3 | - | yes |
| 2MN2 | 29.528 479 | yes | - | - | | - | - |
| 2SK2 | 29.917 863 | yes | - | - | | - | - |
| T2 | 29.958 933 | yes | yes | 0.6 | 1.6 | yes | - |
| S2 | 30.000 000 | yes | yes | 11.8 | 18.2 | - | yes |
| R2 | 30.041 067 | yes | yes | 0.7 | 1.8 | - | - |
| K2 | 30.082 137 | yes | yes | 1.1 | 5.0 | yes | - |
| MSnu2 | 30.471 521 | yes | - | - | | - | - |
| MSN2 | 30.544 375 | yes | yes | 1.1 | 1.2 | - | - |
| KJ2 | 30.626 512 | yes | yes | 0.6 | 1.0 | - | - |
| 2SM2 | 31.015 896 | yes | yes | 1.6 | 2.1 | - | yes |
| 2MS2N2 | 31.088 749 | yes | - | - | | - | - |
| SKM2 | 31.098 033 | yes | - | - | | - | - |





| Name | Speed (deg/hour) | 18yr list | 1 year list | Max effect of surge in UK(cm) | Max effect of surge globally (cm) | 1 month list Related | 1 month list Primary |
|---|---|---|---|---|---|---|---|
| Total: | | 115 | 62 | | | 8 | 26 |
| MQ3 | 42.382 765 | yes | - | - | | - | - |
| NO3 | 42.382 765 | yes | yes | 0.5 | 1.6 | - | yes |
| 2MP3 | 43.009 277 | yes | - | - | | - | - |
| M3 | 43.476 156 | yes | yes | 0.1 | 0.7 | - | yes |
| SO3 | 43.943 036 | yes | yes | 1.1 | 2.4 | - | - |
| MK3 | 44.025 173 | yes | yes | 0.7 | 1.6 | - | yes |
| 2MQ3 | 44.569 548 | yes | - | - | | - | - |
| SK3 | 45.041 069 | yes | yes | 0.5 | 2.4 | - | - |
| 2MNS4 | 56.407 938 | yes | - | - | | - | - |
| 3MK4 | 56.870 175 | yes | - | - | | - | - |
| 3MS4 | 56.952 313 | yes | - | - | | - | - |
| MN4 | 57.423 834 | yes | yes | 0.7 | 1.3 | - | yes |
| Mnu4 | 57.496 687 | yes | - | - | | - | - |
| 2MSK4 | 57.886 071 | yes | - | - | | - | - |
| M4 | 57.968 208 | yes | yes | 1.7 | 3.0 | - | yes |
| SN4 | 58.439 730 | yes | yes | 1.0 | 1.4 | - | yes |
| 3MN4 | 58.512 583 | yes | - | - | | - | - |
| MS4 | 58.984 104 | yes | yes | 3.2 | 4.1 | - | yes |
| MK4 | 59.066 242 | yes | yes | 0.7 | 1.9 | - | - |
| 2MSN4 | 59.528 479 | yes | - | - | | - | - |
| S4 | 60.000 000 | yes | yes | 0.8 | 3.1 | - | - |
| SK4 | 60.082 137 | yes | yes | 0.5 | 2.3 | - | - |
| 3MK5 | 71.911 244 | yes | - | - | | - | - |
| M5 | 72.460 261 | yes | - | - | | - | - |
| MSO5 | 72.927 140 | yes | - | - | | - | - |
| 3MO5 | 73.009 277 | yes | - | - | | - | - |
| MSK5 | 74.025 173 | yes | - | - | | - | - |
| 2(MN)S6 | 84.847 668 | yes | - | - | | - | - |
| 3MNS6 | 85.392 042 | yes | - | - | | - | - |
| 4MK6 | 85.854 280 | yes | - | - | | - | - |
| 4MS6 | 85.936 417 | yes | - | - | | - | - |
| 2MSNK6 | 86.325 801 | yes | - | - | | - | - |

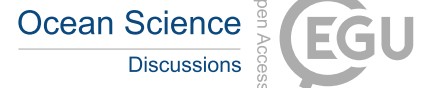



| Name | Speed (deg/hour) | 18yr list | 1 year list | Max effect of surge in UK(cm) | Max effect of surge globally (cm) | 1 month list Related | 1 month list Primary |
|---|---|---|---|---|---|---|---|
| Total: | | 115 | 62 | | | 8 | 26 |
| 2MN6 | 86.407 938 | yes | yes | 0.4 | 0.8 | - | yes |
| 2Mnu6 | 86.480 792 | yes | - | - | | - | - |
| 3MSK6 | 86.870 175 | yes | - | - | | - | - |
| M6 | 86.952 313 | yes | yes | 0.7 | 1.2 | - | yes |
| MSN6 | 87.423 834 | yes | yes | 0.6 | 1.1 | - | yes |
| 4MN6 | 87.496 687 | yes | - | - | | - | - |
| 2MS6 | 87.968 208 | yes | yes | 1.4 | 2.6 | - | yes |
| 2MK6 | 88.050 346 | yes | yes | 0.3 | 0.9 | - | - |
| 3MSN6 | 88.512 583 | yes | - | - | | - | - |
| MKL6 | 88.594 720 | yes | - | - | | - | - |
| 2SM6 | 88.984 104 | yes | yes | 0.6 | 1.4 | - | yes |
| MSK6 | 89.066 242 | yes | yes | 0.4 | 1.5 | - | - |
| 2(MN)8 | 114.847 668 | yes | - | - | | - | - |
| 3MN8 | 115.392 042 | yes | - | - | | - | - |
| M8 | 115.936 417 | yes | - | - | | - | - |
| 2MSN8 | 116.407 938 | yes | - | - | | - | - |
| 3MS8 | 116.952 313 | yes | - | - | | - | - |
| 3MK8 | 117.034 450 | yes | - | - | | - | - |
| MSNK8 | 117.505 971 | yes | - | - | | - | - |
| 2(MS)8 | 117.968 208 | yes | - | - | | - | - |
| 2MSK8 | 118.050 346 | yes | - | - | | - | - |
| 4MS10 | 145.936 417 | yes | - | - | | - | - |
| 3M2S10 | 146.952 313 | yes | - | - | | - | - |
| 4MSN12 | 174.376 146 | yes | - | - | | - | - |
| 5MS12 | 174.920 521 | yes | - | - | | - | - |
| 4M2S12 | 175.936 417 | yes | - | - | | - | - |





*Author contributions.* Williams carried out the model runs and post-processing, using Irazoqui Apecechea's recent developments to the GTSM model code and global grid, led by Verlaan. Saulter advised on Met Office procedures. Williams prepared the manuscript with contributions from all co-authors.

*Competing interests.* The authors declare that they have no conflict of interest.

5   *Acknowledgements.* We are grateful for funding from the EU under the Atlantos project, Horizon 2020 Grant No. 633211, and from the Met Office. Some of the results in this paper first appeared as an internal NOC report Williams et al. (2018).



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
