# Peer review of "Radiational Tides: their double-counting in storm surge forecasts and contribution to the Highest Astronomical Tide."

_Ocean Science, 2018_

## Referee Comment (RC1) · Anonymous Referee #1 · 27 Jun 2018

The manuscript addresses current practices to predict water level changes for various operational marine applications. Such predictions need to include sea surface height changes due to all acting processes including atmospherically induced surge and lunisolar ocean tides. Typically, information on tidal effects and the time evolution of the general ocean circulation are obtained from different sources and are added by means of linear superposition. This assumption of linearity can be, however, questioned in view of distinct periodicities in the atmospheric-induced circulation associated with the either the seasonal cycle or atmospheric tides.

The present manuscript addresses this topic in a way I certainly believe worth to be

published in Ocean Sciences. However, a number of points raised below might be addressed in order to increase clarity of the representation and expand the discussion to cover all relevant aspects of the topic:

(1) The nomenclature applied is somewhat problematic: M obviously represents modelbased sea surface heights, W apparently stands for observed water levels. H, however, is used for harmonic estimations/predictions for either models, observations, or final combined forecasts. This is difficult to comprehend, so I suggest to reserve capital letters to identifying the source of water level informations (i.e., model (M); tide gauge (G); water level forecasts (W)), and indicate the actual signal component by subscripts (time series of tides (t), harmonic estimates from a time series of tides (th); time series of surge and other meteorological forcings (s); harmonic estimates from a time series of tides and surge and other meteorological forcing (tsh), etc.).

(2) W\_g is apparently not properly introduced at all.

(3) The example of Section 2.4 is only partially convincing. What is the usual base period taken to estimate  $H_g$ ? Isn't it plausible to assume that surge event effects on  $H_g$  will cancel out over time? Are there recommendations available on the number of constituents to be considered? What about the treatment of minor tides?

(4) The effects of the annual tide Sa and the semi-annual tide Ssa might be discussed in more detail, in particular in view of the fact that the ocean circulation might have also a distinct annual periodicity.

(5) Changes in river discharge and their consequences on local water levels might be not relevant for the U.K. but can have a profound impact for estuaries in other parts of the world. A few comments about this process might be helpful.

A few rather minor points might be also addressed during the revision:

(5) It could be mentioned somewhere in the text that M2 is also having a very weak atmospheric pressure signature (see 10.1002/2015JD024243).

OSD
(6) Figure 1 is difficult to read. Maybe enlarging the vertical extend of the figure would help?

(7) The frequent change in units between cm and m in the text is rather unfortunate.

OSD

---

## Referee Comment (RC2) · Anonymous Referee #2 · 18 Jul 2018

General Comments

The paper covers a topic that is interesting scientifically and important for storm surge modelling and forecasting, in particular for forecasts systems where the surge plus tide-surge interaction are added to tidal predictions. The paper describes the magnitude of errors that may arise from different processes omitted from some forecasting systems. The methodology is clear and valid. I recommend accepting the paper for publication following minor revisions, which are mostly structural and grammatical.

Specific Comments

The title implies the primary focus for the paper is the effect of radiational tides on surge

forecasts, but the paper covers a number of considerations for storm surge forecasting and navigational chart datums (LAT, HAT). There are inconsistencies in the 'message', for example in the abstract and headings discussing HAT, when in fact a discussion of HAT and LAT is made. I recommend amending the title to cover the full content of the paper; for example "Errors arising from the treatment of radiational tides in storm surge forecasting and tide-based datums". Also, given the structure of the paper follows a report style, a walk through the paper structure at the end of the Introduction (Page 2, Ln 11) would be very useful.

The numerical model is forced by ECMWF ERA-Interim wind fields with a resolution of 6 hours. It is my understanding that the storm surge numerical model will therefore lack some of the 'peakiness' in surge and high-frequency oscillations in the modelled tide+surge total water level (Ws) compared with tide gauge observations (at hourly or higher frequency sampling; Wg). Can the authors comment on the effective frequency of the surge signal by their use of a numerical model and can they quantify the effect on tide magnitude and phase estimation, e.g. quarter-diurnal shallow-water tidal constituents? I would imagine, since the numerical model will underestimate the total power in the signal, versus observations, that the double-counting of meteorological effects in the harmonic prediction are even larger than presented here. This extra work is not necessary for publication but would be interesting.

Pg 5 ln 8: Please explain what Byrne and Flowerdew were pointing out, and hence why this fortnightly periodic error is important.

Minor typographical and grammatical notes

Please be consistent with "tide gauge", "storm surge" or "gauge", "surge".

Please be consistent between numerical model run labels "surge+tide" and "tide-only" (there are many references to "surge" which could be mis-understood as surge-only as the phraseology is not clearly introduced in the Introduction).

Pg 1 Ln 3: Suggest "In some storm surge forecasting systems, a regional model is run twice: once as tide-only, . . ., and again as tide-and-surge, ... "

Pg 1 Ln 8: Suggest change "key constituents" to "major constituents"

Pg 1 ln9/10: Suggest the authors emphasise why HAT and LAT are important. Suggest change "We also quantify the extent to which the Highest Astronomical Tide, which is derived from..." to " We also quantify the extent to which tide levels used in navigation datums and design heights, the Lowest and Highest Astronomical Tides (which are derived ...)"

Pg 2 ln 4: Please reference Appendix A; else there is no citation or reference to the GSTM development and version in the main text.

Pg 2 Chapter 2: The notation is quite confusing. A notation table as an Appendix would be useful, clarifying what denotes total water level, tide and surge from what denotes numerical model or tide gauge observations and harmonic predictions.

For much of Chapter 2, the authors are clearly discussing the UK system. Can you make it clear 'we use' is specifically referring to the operational system in the UK. Where a methodology is typically followed by the sea level community, make that clear; for example, on Page 2 Ln 29+, "The choice and number of tidal constituents determined by harmonic analysis are typically chosen according to the length and frequency of data available"

Pg 7 Figure 4. Cyan line with baseline at 2.2 m has no label (total?), please check the labels.

Pg 7 ln 1: Change "effected" to "affected"

Pg 8 ln 2: Change "Vector differences . . . is . . ." to "A vector difference of . . . in S2 is . . ."

Pg 10 Chapter 4. Change heading to also include Lowest Astronomical Tide, or simply
refer to tide-based datums

Author Contributions refer to Verlaan, who is not an author, so perhaps provide institution for completeness/clarity.

---

## Editor Comment (EC1) · R. D. Ray (Editor) · 27 Jul 2018

Dear Dr Williams and colleagues:

Since all formal (and informal) reviews are now in, I invite you to address the reviewer comments and to revise your paper accordingly. Let me also add a few points to those made by reviewers:

1. At several points in the paper, you refer to the nodal tide (or node tide). I think most readers will take this to mean the near-equilibrium zonal tide of period 18.6 years. But surely you are instead referring to 18.6-y modulations of all lunar tides (especially the

large ones like M2, O1, and K1). Yes? I think Phil, in informal comments, also thought you were referring to the 18.6-y tide; there is thus no reason to cite his 2012 paper on the topic, as it's irrelevant (assuming I'm right about what you've meant). So if you really mean nodal modulations of major tides, it's best not to call that the node tide, even though all of this does arise from the moon's nodal precession.

2. Both reviewers found the nomenclature problematic – e.g. see Point (1) of Reviewer #1 – and I also had to repeatedly read the relevant text because I kept getting confused about what was what. So please give some thought to making this clearer, possibly along the lines suggested by Reviewer #1, or some other way if you have a good idea.

3. Many figures are difficult to see because they are so small, and their fonts are even smaller. Remember that most journals end up reducing figure sizes anyway, so give some care to figure legibility.

---

## Author Comment (AC1) · 2 Aug 2018

**Thank-you for your helpful review. Please see inline replies.**

The manuscript addresses current practices to predict water level changes for various operational marine applications. Such predictions need to include sea surface height changes due to all acting processes including atmospherically induced surge and lunisolar ocean tides. Typically, information on tidal effects and the time evolution of the general ocean circulation are obtained from different sources and are added by means of linear superposition. This assumption of linearity can be, however, questioned in view of distinct periodicities in the atmospheric-induced circulation associated with the

either the seasonal cycle or atmospheric tides. The present manuscript addresses this topic in a way I certainly believe worth to be published in Ocean Sciences. However, a number of points raised below might be addressed in order to increase clarity of the representation and expand the discussion to cover all relevant aspects of the topic:

(1) The nomenclature applied is somewhat problematic: M obviously represents modelbased sea surface heights, W apparently stands for observed water levels. H, however, is used for harmonic estimations/predictions for either models, observations, or final combined forecasts. This is difficult to comprehend, so I suggest to reserve capital letters to identifying the source of water level informations (i.e., model (M); tide gauge (G); water level forecasts (W)), and indicate the actual signal component by subscripts (time series of tides (t), harmonic estimates from a time series of tides (th); time series of surge and other meteorological forcings (s); harmonic estimates from a time series of tides and surge and other meteorological forcing (tsh), etc.).

Thank-you for the suggestion. All reviewers commented on the notation, and we've completely revised it. Rather than the double subscript, we've changed to an overhead tilde to indicate "time series derived from harmonics", as the shape is reminiscent of a sine wave. We also switch to F for "Forecast water level" and G for (gauge) observed total water level.

Then the forecast is given by  $F = (M_s - M_t) + \tilde{G}$ , the harmonic prediction derived from the tide-and-surge model is  $\tilde{M}_s$ , etc.

(2)  $W_q$  is apparently not properly introduced at all.

The observed total water level. Corrected alongside all notation.

(3) The example of Section 2.4 is only partially convincing. What is the usual base period taken to estimate  $H_g$ ? Isn't it plausible to assume that surge event effects on  $H_g$  will cancel out over time? Are there recommendations available on the number of constituents to be considered? What about the treatment of minor tides?

Section 2.4 (on non-linearity, now 2.5) is about the effect on prediction of an individual surge event, given that there exists some discrepancy in phase between model tides  $M_t$  and the harmonic prediction from the gauge  $\tilde{G}$ .

If during such an event the surge causes an advancement of the tide, then  $M_r = M_s - M_t$  is decreasing rapidly during High Water (the peak of  $M_t$ ). The peak of  $M_r + \tilde{G}$  is therefore dependent on the relative timing of the peak of  $\tilde{G}$  and  $M_t$ .

Thank-you for the prompt to look at this again, as it turns out that for a simple tide it can be shown analytically thus (added to section 2.5):

We construct an example with model tide  $M_t = A\cos(\sigma t)$ , and a surge in which there is an additional uniform water level  $A_s$  and an advancement of the tide of  $t = \delta$ , so model surge is  $M_s = A_s + A\cos(\sigma(t + \delta))$ . The model residual is given by  $M_r = M_s - M_t$ .

Suppose the harmonic prediction at the gauge has the same amplitude but the tide is  $\epsilon$  ahead of the tide-only model,  $\tilde{G} = A \cos(\sigma(t + \epsilon))$ .

The forecast water level is  $F = M_r + \tilde{G}$  and the error in the skew surge forecast is  $\max(F) - \max(M_s)$ . Substituting in and assuming phase changes are small, the skew surge forecast error is

$$E = A \left( \cos(\sigma \epsilon) - \cos(\sigma(\delta + \epsilon)) + \cos(\sigma \delta) - 1 \right).$$

Suppose for example that A = 3 m,  $\sigma = 2\pi/12.42$ , the difference between model and gauge tide is  $\epsilon$ =0.083 (5 min for M2), and the surge advances the tide by 30 min ( $\delta = 0.5$ ). Then E = 0.03 m. This is still below the level of other forecast errors at the moment, but may not be negligible in future.

Although in practice there are many more constituents than M2, a similar relationship will hold in a small window about high tide, with a changing amplitude each day. Indeed, the absence of a small constituent will often manifest as a small phase change in M2. If there are frequent surges, we would expect  $\epsilon$  to have the same sign as  $\delta$ , as the gauge

СЗ

would register water levels more like the surge+tide model than the tide-only model and the harmonic predictions would follow suit.

(4) The effects of the annual tide Sa and the semi-annual tide Ssa might be discussed in more detail, in particular in view of the fact that the ocean circulation might have also a distinct annual periodicity.

There are several contributions to annual cycles that we have omitted from this study, including steric effects, circulation changes, river input, ocean mass changes, gravitational changes... the larger of these are explicitly noted in the introduction. However since these are omitted from both model runs (other than very small effects via the atmosphere) they are not at risk of double counting.

We have added some notes on this in the introduction, and also expanded a little on the results seen for Sa. These in large part follow the local annual cycle in atmospheric pressure, an exception being in the wind-dominated Baltic.

(5) Changes in river discharge and their consequences on local water levels might be not relevant for the U.K. but can have a profound impact for estuaries in other parts of the world. A few comments about this process might be helpful.

Good point. It's a reason that using the tide gauge prediction rather than just model tides may be necessary. Added:

There are other contributors to water level, including steric effects and river flow, that will also create differences between the tide gauge and the forecast water levels, particularly seasonally. The problem of double-counting of periodic changes does not arise if they are omitted from the surge model entirely, but they may contribute to HAT and LAT calculations. These effects are not considered in this study.

A few rather minor points might be also addressed during the revision: (5) It could be mentioned somewhere in the text that M2 is also having a very weak atmospheric pressure signature (see 10.1002/2015JD024243).

Thank-you for the suggestion. Schindelegger 2016 was a very confusing paper till I realised they were using L2 to mean M2 in the atmosphere, when L2 is used in tidal analysis for a subtle lunar elliptical effect at a different frequency! Note added in first paragraph of section 3.

"There is a very small atmospheric tide at  $\mathbf{M}_2,$  peaking at the equator at about 0.1 mbar [Schindelegger 2016] . "

(6) Figure 1 is difficult to read. Maybe enlarging the vertical extend of the figure would help?

Now enlarged as much as possible.

(7) The frequent change in units between cm and m in the text is rather unfortunate.

The figures all use metres. Text is now consistently cm except where refering to absolute tidal heights, HAT and water levels which are of the order of m.

---

## Author Comment (AC2) · 2 Aug 2018

Thank-you for your helpful review. Please see inline comments

*General Comments*

*The paper covers a topic that is interesting scientifically and important for storm surge modelling and forecasting, in particular for forecasts systems where the surge plus tidesurge interaction are added to tidal predictions. The paper describes the magnitude of errors that may arise from different processes omitted from some forecasting systems. The methodology is clear and valid. I recommend accepting the paper for*

*publication following minor revisions, which are mostly structural and grammatical.*

*Specific Comments*

*The title implies the primary focus for the paper is the effect of radiational tides on surge forecasts, but the paper covers a number of considerations for storm surge forecasting and navigational chart datums (LAT, HAT). There are inconsistencies in the 'message', for example in the abstract and headings discussing HAT, when in fact a discussion of HAT and LAT is made. I recommend amending the title to cover the full content of the paper; for example "Errors arising from the treatment of radiational tides in storm surge forecasting and tide-based datums".*

We have amended the title to

"Radiational Tides: their double-counting in storm surge forecasts and contribution to the Highest Astronomical Tide."

Including LAT in the title as well felt a bit clumsy, but it is now explicitly in section headings.

*Also, given the structure of the paper follows a report style, a walk through the paper structure at the end of the Introduction (Page 2, Ln 11) would be very useful.*

Added some links to specific sections and a little more detail is given about the connection to HAT.

*The numerical model is forced by ECMWF ERA-Interim wind fields with a resolution of 6 hours. It is my understanding that the storm surge numerical model will therefore lack some of the 'peakiness' in surge and high-frequency oscillations in the modelled tide+surge total water level (Ws) compared with tide gauge observations (at hourly or higher frequency sampling; Wg). Can the authors comment on the effective frequency of the surge signal by their use of a numerical model and can they quantify the effect on tide magnitude and phase estimation, e.g. quarter-diurnal shallow-water tidal constituents? I would imagine, since the numerical model will underestimate the total*

*power in the signal, versus observations, that the double-counting of meteorological effects in the harmonic prediction are even larger than presented here. This extra work is not necessary for publication but would be interesting.*

There is more detail about the effect of the 6-hourly forcing on capturing surges in the model validation paper (Irazoqui et al 2018, in review). With the 6-hourly forcing there is some underprediction of surges due to tropical cyclones which is improved by the ERA-5 reanalysis (available too recently for the model runs carried out in this paper.) However, it should not contradict the main results a great deal, since tropical cyclones at any given location are sufficiently rare that the tidal coefficients fitted over a year should not be very different if the surges are slightly underestimated. A note on this has been added to the description of the model. "We make the assumption that tropical cyclones at any given location are sufficiently rare that the tidal coefficients fitted over a year should not be very different if the surges are slightly underestimated."

*Pg 5 ln 8: Please explain what Byrne and Flowerdew were pointing out, and hence why this fortnightly periodic error is important.*

They both observed a similar error in forecast high-water levels compared to observations. It is very clear in the Byrne report, but unfortunately that is only in the grey literature.

*Minor typographical and grammatical notes*

*Please be consistent with "tide gauge", "storm surge" or "gauge", "surge".*

It is now consistently the longer form on first usage in a section and the shorter form is used for brevity where no confusion is likely.

*Pg 2 Chapter 2: The notation is quite confusing. A notation table as an Appendix would be useful, clarifying what denotes total water level, tide and surge from what denotes numerical model or tide gauge observations and harmonic predictions.*

The notation is now changed to be clearer (see response to review 1) and hopefully

this is now not necessary.

*For much of Chapter 2, the authors are clearly discussing the UK system. Can you make it clear 'we use' is specifically referring to the operational system in the UK. Where a methodology is typically followed by the sea level community, make that clear; for example, on Page 2 Ln 29+, "The choice and number of tidal constituents determined by harmonic analysis are typically chosen according to the length and frequency of data available"* Reworded. "Similar procedures are implemented elsewhere in the world, as noted above, so in this paper we replace the shelf model with GTSM to examine results globally."

All other minor notes are accepted and corrected accordingly.

---

## Author Comment (AC3) · 2 Aug 2018

Thank-you for your helpful editorial comments and the discussion with Phil.

*1. At several points in the paper, you refer to the nodal tide (or node tide). I think most readers will take this to mean the near-equilibrium zonal tide of period 18.6 years. But surely you are instead referring to 18.6-y modulations of all lunar tides (especially the large ones like M2, O1, and K1). Yes? I think Phil, in informal comments, also thought you were referring to the 18.6-y tide; there is thus no reason to cite his 2012 paper on the topic, as it's irrelevant (assuming I'm right about what you've meant). So if you really mean nodal modulations of major tides, it's best not to call that the node tide,*

[Figure]

*even though all of this does arise from the moon's nodal precession.*

Yes, that's right, I should have referred to "nodal modulation". Now corrected throughout.

*2. Both reviewers found the nomenclature problematic – e.g. see Point (1) of Reviewer #1 – and I also had to repeatedly read the relevant text because I kept getting confused about what was what. So please give some thought to making this clearer, possibly along the lines suggested by Reviewer #1, or some other way if you have a good idea.*

We've revised the notation, and hopefully it's less confusing now, with tilde to indicate "harmonic predictions from..." See response to reviewers.

*3. Many figures are difficult to see because they are so small, and their fonts are even smaller. Remember that most journals end up reducing figure sizes anyway, so give some care to figure legibility.*

The figure fonts are now increased to be similar to the main text, and figures are enlarged.

As you know, we also received a very detailed review from Phil Woodworth. Most of his comments were on detailed presentation and precise wording, and they have all been addressed, with the exception of the choice of named coastal locations. These were a compromise between even spacing and well known places, and hopefully the clearer legend and reference to the map in the Appendix will make these figures easier to understand.

There was a discussion about the relative importance of the nodal tide in tidal range and the paragraph in section 4 now reads:

An approximate calculation of $\mathrm{Range} = 2(\mathbf{M}_2 + \mathbf{S}_2 + \mathbf{O}_1 + \mathbf{K}_1)$ is occasionally used [Yotsukuri 2017], but the error due to this can be over 1 metre (figure 6b). $\mathbf{N}_2$ is a significant contributor, at about 20% of $\mathbf{M}_2$ in many sites worldwide. A few tens of centimetres are accounted for by the omission of the nodal modulations, and there are

also the shallow water constituents at the coast.

---

## Author Response (AR2)

**1 Response to editor minor revisions**

Thank-you Richard. I've addressed all you comments below,

*Topic Editor Decision: Publish subject to minor revisions (review by editor) (15 Aug 2018) by Richard Ray Comments to the Author: Dear Jo & colleagues:*

*Co-editor Phil Woodworth and I have both gone through the revised paper and compared reviewer comments and replies. Before final acceptance, there are a few items that need to be addressed. Most are very minor.*

*The paper is now much clearer, with the new notation being especially helpful. In fact, I recommend that you add somewhere the point you make in the Replies, that a tilde denotes a harmonic prediction – otherwise, a reader might fail to notice that helpful guide.*

done

*Given that I now understand the paper far better than I initially did, I have a couple of more substantial points (or puzzlements).*

*1) Page 5, line 20 and surrounding: You express mild surprise that tide+surge has large perturbations of Mm and Mf (and also MSf, although you note that could be from M2-S2 interaction). One possible explanation that occurs to me is the following: With atmospheric forcing, the sea level response turns very red, putting lots of non-tidal power into periods that will get soaked up in Mm and Mf estimates. This is why reliable tide-gauge estimates of Mf,Mm almost always require a many-year time series.*

*I'm actually far more surprised by the large effects at M2.*

Yes. I've added a note that: " Another possibility is that non-tidal power in the tide-and-surge model is leaking into $\mathbf{M}_\mathrm{m}$ and $\mathbf{M}_\mathrm{f}$ estimates. Eliminating this would require a many-year model run."

I think the M2 change comes from the consistent bias due to surges slightly altering the tide in the same way in a given location and pushing a bit of energy into one of the shallow water terms. I'm not absolutely sure though. We had some nice model results from another project that might help clear it up - details may appear in another paper!

*2) I am puzzled why the tide prediction errors in Fig 1b are so large. This is just $\tilde{M}_t$ being computed from harmonics directly estimated from $M_t$ and yet 62 constituents fail to capture so much of the signal?? And by my eye, for some locations, the error appears to be fortnightly (i.e. spring-neap periodicity) and more or less constant throughout the year.*

*I'm assuming that the 62 harmonic constants were computed for the 2013 timespan, identical to the timespan used for the figure. True? Or are the harmonics computed from a different timespan, as in Flowerdew et al (2010)? If the latter, I can understand this more readily.*

*In any event, it would be interesting to understand how these errors arise. For some location showing this large error (perhaps 10 cm?), it may be enlightening to examine in more detail why the 62 harmonics leave so much error.*

*You may argue that such additional research is beyond the scope of the present paper, and I accept that. In fact, it would likely require more than a single year of data to understand what is going on. However, understanding*

*this error should be important to your group. So I think further analysis is warranted at some point.*

You are correct that the harmonics are derived from the same period, and yes there are sites where 62 do not represent even the tide-only model well. It's better when you go to the 115 list.

Eg around Avonmouth, with 62 and 115 constituents:

[Figure]

This is something I absolutely do want investigate in more detail - to understand better how to select constituents around the world, when it is necessary to use a very large number, and how the errors are affected by the length/density/noise characteristics of the time series. But I think we have to leave it for another paper.

Edited the paragraph describing that figure to include: "In practice $\Delta <$ 5 cm at most UK sites and the monthly cycle has gone, but in the Bristol Channel there is still an error of around 50 cm, indicating that the 62 harmonic constituents are not capturing all of the model tide, and further shallow-water constituents may be required."

*3) Page 4, line 1: with the change in notation, W should be G ?*

Yes. Corrected.

*More minor points: 4) Page 6, with the change in notation, do you still want to use H(t) for this section?*

No, that should be $\tilde{G}(t)$. Corrected.

*5) Section 3.1: Minor point, but not only does ECMWF contain the S2 air tide, but it's an incorrect representation of it. This is because 6-h sampling is the S2 Nyquist, so ECMWF cannot capture the full S2 signal. The point is made in papers on air tides, although not very germane in the present context, so you can mention or not as you see fit.*

Added:

" The 6-hour sampling prevents ERA-Interim forcing from capturing the $S_2$ atmospheric tide correctly [Dobslaw2005], but the analysis in this paper is self-consistent with the forcing used."

*6) Figure 3, right panels. I cannot distinguish the red lines. Are they BOTH supposed to be dashed? Compare to left panels.*

Corrected. I've also tidied up this figure so it'll fit in single column.

*7) Page 3, line 2. New text: "we replace the shelf model". WHAT shelf model? None has previously been mentioned up to this point, other than those of other groups.*

It was in the previous paragraph. Slightly rephrased to tie better.

*8) The figure legibility is mostly better now. I see you enlarged fonts in Fig 6b, but not 6a. Moreover, the color scale is cut off at the high end on both panels.*

Edited.

*Phil Woodworth also has the following comments. Note especially his point about the 115 constituents, which if he (and I) understand the paper correctly would allow you to reduce somewhat the very large Table B1.*

*p3, 6 - define ECMWF acronym*

done

*21 - maybe I have missed something, but she has no real tide gauge data, only the 1 year and 1 month GSTM data sets mentioned in lines 11-12. So what is the '115 for more than one year' relevant for here? I could not see a mention of using this set in the rest of the paper. I had noticed this last time but for some reason it didn't make my list. if this is dropped then it means removing the 115 set from the appendix B table.*

I did do some tests that the results detailed at the beginning of section 3 weren't too dependent on the choice of constituents. But in the interests of clarity I've tidied table B1 to only be the 1 year /1 month lists, and added a supplementary spreadsheet with the 115.

*p5, 32 - funny that MA2 should change so much and MB2 doesn't. But no matter.*

It's only a bit less that the cut-off I chose.

*p6, 18 - I would reword this slightly as it is a bit cryptic and I was puzzled what 'this data' meant for a while:*

*Since the GSTM data used for this was the 1 month set (Jan 2012), this exercise made use of only ...*

*I don't understand why they used the 1 month set and not the 1 year set for this, all they are doing is deriving some harmonic constants which are then run for 18.6 years*

It's because I wanted the full map, and I decided only to save the high-frequency time series every grid point for 1 month because of the amount of data. In hindsight I'd use a sparse grid everywhere for output. There's also the one-year run processed on the coast. Rephrased to clarify.

*p14, 3 - GESLA needs referencing, either to http://www.gesla.org and/or the journal reference:*

*Woodworth, P.L., Hunter, J.R. Marcos, M., Caldwell, P., Menendez, M. and Haigh, I. 2017. Towards a global higher-frequency sea level data set. Geoscience Data Journal, 3, 50-59, doi:10.1002/gdj3.42.*

Apologies! Rectified.

*p15, figure A1 - this is much better than before and I even understand why the Bering St is where it is in the list now. Could the unphysical 100 and -100 be dropped from the labelling on the y-axis? Doesn't matter much I guess. I thought at first the dashed line at -90 was simply indicating the pole but there isn't one at +90, and there are 3 other dashed lines at Europe latitude. the caption should say what they are.*

The dashed lines were left over from an earlier figure explaining the construction but they don't add anything here. Edited.